

# Robust and automatic definition of microbiome states

Beatriz García-Jiménez and Mark D. Wilkinson

Centro de Biotecnología y Genómica de Plantas UPM-INIA, Universidad Politécnica de Madrid, Madrid, Spain

## ABSTRACT

Analysis of microbiome dynamics would allow elucidation of patterns within microbial community evolution under a variety of biologically or economically important circumstances; however, this is currently hampered in part by the lack of rigorous, formal, yet generally-applicable approaches to discerning distinct configurations of complex microbial populations. Clustering approaches to define microbiome "community state-types" at a population-scale are widely used, though not yet standardized. Similarly, distinct variations within a state-type are well documented, but there is no rigorous approach to discriminating these more subtle variations in community structure. Finally, intra-individual variations with even fewer differences will likely be found in, for example, longitudinal data, and will correlate with important features such as sickness versus health. We propose an automated, generic, objective, domain-independent, and internally-validating procedure to define statistically distinct microbiome states within datasets containing any degree of phylotypic diversity. Robustness of state identification is objectively established by a combination of diverse techniques for stable cluster verification. To demonstrate the efficacy of our approach in detecting discreet states even in datasets containing highly similar bacterial communities, and to demonstrate the broad applicability of our method, we reuse eight distinct longitudinal microbiome datasets from a variety of ecological niches and species. We also demonstrate our algorithm's flexibility by providing it distinct taxa subsets as clustering input, demonstrating that it operates on filtered or unfiltered data, and at a range of different taxonomic levels. The final output is a set of robustly defined states which can then be used as general biomarkers for a wide variety of downstream purposes such as association with disease, monitoring response to intervention, or identifying optimally performant populations.

## INTRODUCTION

The study of microbiome dynamics is extremely important (*Gilbert et al., 2016*; *Bashan et al., 2016*; *Bradley & Pollard, 2017*), but is currently hindered by the lack of tools that can reliably detect statistically distinct states within microbial populations, and categorize new populations *vis-a-vis* these states. Microbial communities differ between different environments, but also differ within the same environment over time (*Bashan et al., 2016*). One widely-accepted view of the microbiome within a given environment is to consider it

Corresponding author
Beatriz García-Jiménez,
beatriz.garcia@upm.es

from a global perspective: among all individuals there is one shared state, broadly-defined, but internally variable and dynamic (*Gibbons et al., 2017*). With respect to gut microbes, the research community initially attempted to associate distinct microbial population structures with lifestyle, culture, or diet - called "enterotypes". However, there is now much doubt about the existence of enterotypes, and if they exist, how they are defined (*Jeffery et al., 2012*; *Costea et al., 2018*). Nevertheless, even if there are no universally-distinct enterotypes, this does not preclude the possibility that microbial populations may exist in distinct configurations that are stable in the long or even short-term, and that these are associated with important traits such as health (*Gilbert et al., 2016*; *Woloszynek et al., 2016*; *Shankar, 2017*). Indeed, studies point to the existence of such stable steady-states, both from experimental data (*Turroni et al., 2017*) and from modeling approaches (*Stein et al., 2013*; *Bashan et al., 2016*) where "steady-state" refers to the average microbial composition over a period of time, rather than a precisely defined composition (*Chan, Simons & Maranas, 2017*).

*Faust et al. (2015)* reported that microbial diversity was relatively stable over time, in a stable environment, but that stability could be disrupted by (a) external perturbations, (b) direct modifications or (c) transient perturbations, all of which cause the microbial community to change. These changes may be gradual (*MacDonald, Parks & Beiko, 2012*; *Gibbons et al., 2017*) or step-wise and discrete (*Zhou et al., 2014*; *Turroni et al., 2017*). Even subtle population changes may have significant external consequences; however associating biologically or economically important features with specific microbial population configurations requires robust and reproducible approaches to defining microbial population configurations. Flexible, domain-independent tooling that enables the rigorous definition and detection of microbial population states are vital to enable the study of microbiome evolution and dynamics, and the consequences of interventions or perturbations.

In this manuscript, we define a microbiome "state" as a collection of constraints satisfied by the microbiota of one set of samples, that are not satisfied by other samples, allowing them to be reliably distinguished from one another. That is to say, our state definitions are purely data-driven, without regard to any biological features or considerations at the individual or community level, and are defined on a dataset-by-dataset basis, making no claims about individuals or populations outside of that dataset. We consider microbiome states to be biomarkers which may differ between individuals and/or within the same individual over time and, in some cases, may act as "a measurable indicator of a biological state or environmental exposure" (*Gorvitovskaia, Holmes & Huse, 2016*). Thus, from this perspective, a given microbial community could move through different states over time, notwithstanding any overarching definition of that community's "enterotype" as defined by *Arumugam et al. (2011a)*, or "community state type" as defined by *Gajer et al. (2012)*. While the availability of an objective, robust, and widely-applicable approach to defining microbiome state would have great utility, not least by providing a common framework for comparing microbiome studies to one another and for doing association studies with biological phenomenon of interest, our primary interest in creating this algorithm was its application in the context of longitudinal studies. To contribute to the field of microbiome
dynamics, we required an algorithm capable of defining states outside of (and in addition to) the typical static context. Numerous previous studies involved grouping microbiomes of multiple subjects, assuming their microbiomes are largely similar and stable (*Gonze et al., 2017*; *Costea et al., 2018*). However, in the context of microbiome dynamics, we must detect what may be relatively small changes in same individual over time. There are relatively few studies of temporal microbiome dynamics that specifically highlight different states within any microbiome site (*Gajer et al., 2012*; *Ding & Schloss, 2014*; *David et al., 2014*; *Lahti et al., 2014*; *Bucci et al., 2016*; *Baksi, Kuntal & Mande, 2018*; *García-Jiménez, De la Rosa & Wilkinson, 2018*). For those that do, the approach to microbiome sample clustering is varied and somewhat *ad-hoc*. This highlights the fact that the definition of intra-individual microbiome states is non-trivial (*Koren et al., 2013*). Thus, we consider creation of a generally-applicable algorithm that defines and detects robust, individual-level microbiome states within any input dataset a beneficial contribution to the community—one that will aid in both the pursuit of novel studies, as well as the cross-replication and comparison between studies.

The inputs to this algorithm are Operational Taxonomic Unit (OTU) vectors, where each OTU represents a group of species considered indistinguishable by the OTU grouping process, and whose abundance in a set of samples has been computed. Given that these abundance distributions are effectively continuous - i.e., the space of the OTU vectors is so large as to be computationally intractable - the set of OTU combinations that could be defined as representative of a microbial population state must be simplified. However, increasing simplification results in a decreasing number of states, eventually resulting in states with insufficient granularity to occur in any "biologically meaningful" association. This is therefore the key consideration when defining the approximation for grouping similar OTU vectors by, for example, machine learning clustering approaches. The choice of clustering parameters should be guided by the desire to identify several well-populated microbiome states, both within and between individuals, which can then be used as the basis of a model associating these states with biologically interesting phenomenon. It should be noted that this is a distinct goal from the large majority of microbiome studies, where the objective is to create an inclusive and broad definition of the microbiome, absorbing and masking any within- or between-individual differences. As such, the default clustering parameters that appear in most published approaches do not match our problem requirements, and must be considered from scratch. Our approach to defining states, described in detail below, consists of applying a clustering algorithm to the microbime data, and taking a metagenomics beta diversity metric as the distance measure between samples. We then attempt to robustly define the optimal number of clusters based on a comparison between several distance measures, distinct algorithms and different clustering scores.

Metagenomics sample clustering has been achieved for different studies using a variety of approaches, in terms of distance measure, algorithm and number of clusters. For example, as a distance measure, the Jensen-Shanon Distance (JSD) (or its root squared, rJSD, as in (*Arumugam et al., 2011a*)) is the most frequently used (*Gajer et al., 2012*); although the cophenetic or the Euclidean distance are also sometimes applied. Several clustering

algorithms have been used to group metagenomics samples, such as PAM, Agnes, Hclust, or Dirichlet Multinomial Mixture, with different linkage options (*Ding & Schloss, 2014*). For determination of the number of clusters, prior studies have used a diverse array of assessment criteria including the average Silhouette width (SI), Calinski-Harabasz (CH) index, Laplace approximation, etc. With respect to clustering applied to the definition of enterotypes, (*Arumugam et al., 2011a*) provides a tutorial (*Arumugam et al., 2011b*) in which they compute the distance as the root square of the JSD, with the PAM algorithm and selecting the number of clusters with the CH index combined with a SI assessment. Conversely, *Gajer et al. (2012)* applied hierarchical clustering with JSD and SI assessment. It is therefore evident that individual studies may apply widely different, even *ad hoc*, approaches to the identification of groupings in microbiome datasets, and there is little in the way of common standards or workflows; moreover, the outputs are often accepted without further tests of the robustness of the discovered groupings. This hinders inter-study comparisons, and presents a challenge for scholarly reproducibility.

This manuscript describes a multi-step algorithm where the most robust set of "states" (effectively, clusters) is generated for the input microbiome dataset. Briefly, the configurable steps that are combined to generate the robustness of our algorithm include: five different distance measures (JSD, rJSD, Bray-Curtis, Morisita-Horn and Kulczynski), two types of clustering scores (SI and Prediction Strength (PS)) followed by an additional bootstrapping process (evaluated with the Jaccard similarity score), and two distinct clustering approaches (PAM and Hclust).

The utility of the proposed new algorithm is verified through reanalysis of eight previously-published microbiome datasets, where we contribute new insights into the microbiome states discovered within these datasets. The reanalysed datasets have been selected to reflect our study format of interest (longitudinal sampling of individuals); however, the algorithm is generally applicable to any dataset containing a sufficient number of similar samples.

The contribution of this work is an objective and robust mechanism for identifying and tracking distinct states within microbiome datasets in a manner that they can be utilized as biomarkers in, for example, association studies, or exploring how and why microbiome state transitions happen. These could in-turn address the challenge of predictably influencing microbiota dynamics to achieve important objectives such as personalized medicine (*Gilbert et al., 2016*; *Woloszynek et al., 2016*), sustainable agriculture or industrial production (*Valseth et al., 2017*). Indeed, we demonstrate the utility of this algorithm in our recently published MDPbiome study (*García-Jiménez, De la Rosa & Wilkinson, 2018*), where we utilize this novel robust clustering approach to identify biologically relevant states and intra-individual state transitions in the absence of objective expert knowledge, and to predict how interventions will affect state-changes.

## MATERIALS AND METHODS

### Automating the identification of granular, yet robust states

We first provide a brief overview of our automated procedure to define microbiome states to help orient the reader through the more detailed description that follows. Our robust

clustering methodology takes a normalized OTU matrix as input, in the form of either: (a) a phyloseq object (*McMurdie & Holmes, 2013*) or (b) a BIOM format file (*McDonald et al., 2012*), together with their corresponding metadata identifying each sample and taxa.

The clustering procedure is based on: (1) *Koren et al. (2013)*'s study; (2) bootstrapping in clustering; and (3) similarity measures from (*Barwell, Isaac & Kunin, 2015*). *Koren et al. (2013)* recommends testing multiple approaches for the definition of enterotypes, and then comparing results. Following these suggestions, our algorithm first finds clusters using two different methods (PAM and Hclust), applying five different distance metrics (JSD, root-JSD, Bray-Curtis, Morisita-Horn and Kulczynski) each, and nine different seed cluster numbers, $k$ (in the range from 2 to 10). In particular, we adapt the *Koren et al. (2013)* approach to the distinct problem of defining microbiome states that exhibit short-term transitions within the same individual, rather than the long-term stereotypes common to enterotype studies, with sparse transitions and where each individual has just one associated state.

Using the output from this clustering step, we begin to identify the most robust results through a novel assessment approach that utilizes the following criteria:

1. Choosing the $k$ number of clusters (from 2 to 10) with the highest average Silhouette width (SI) among all combinations of pairs of beta diversity measures, with the score being above the SI threshold (0.25), and

2. Checking if that $k$ value also passes the Prediction Strength (PS) threshold (0.80) for robustness, or

3. Confirming if those $k$ clusters are stable according to the Jaccard threshold (0.75) from a bootstrapping process

We apply those criteria to the output of the 90 combinations (2 algorithms $\times$ 5 distances $\times$ 9 $k$ values) per dataset to discard those that are not robust and/or not reproducible.

The final output of our algorithm is a phyloseq object with a new variable defining the cluster identifier into which each sample has been grouped, a file with <sampleID, clusterID > pairs, and a set of clustering assessment graphs (described below) that can be used to judge robustness.

The following sections explain each step in detail, and the relevant factors considered at each point in the automated procedure.

## Clustering approaches

We selected two widely-used algorithms as representative of the two most common clustering approaches: (a) PAM (*Kaufman & Rousseeuw, 1990*), as a partitioning approach, and (b) Hclust (*Kaufman & Rousseeuw, 1990*), as an agglomerative hierarchical approach.

We selected PAM (Partitioning Around Medoids) (*Kaufman & Rousseeuw, 1990*) because it is an improved approach to the well-known non-deterministic $k$-medoids (*Kaufman & Rousseeuw, 1987*). This improvement is achieved in two ways. First, PAM selects $k$ medoids among the input instances in a greedy phase, rather than a random selection of the $k$-medoids (*Reynolds et al., 2006*). The greedy phase takes each new medoid to minimize the objective function, that is, the sum of distances between each instance and its medoid. Therefore, PAM is a deterministic algorithm and does not

need to be run multiple times, as opposed to the *k*-medoids approach. Second, it spreads out the remaining instances among the defined medoids according to the minimum distance-to-medoid criterion, using the values defined in the distance matrix. PAM then selects each possible pair of instances <medoid, not-medoid >, and evaluates whether the swapping between different clusters results in a smaller value for the objective function. This final step is repeated until the set of medoids do not change.

Hclust, a hierarchical clustering algorithm, adopts a *bottom-up* approach (*Kaufman & Rousseeuw, 1990*). Hclust begins with an independent cluster per instance. The two nearest clusters are then grouped, in an iterative way, until the algorithm arrives at a single cluster representing the root of an inverted tree structure. We chose *average* (i.e., UPGMA) distance between cluster members as the linkage criteria used to compute the distance between two clusters, rather than single or complete linkage which takes only one cluster element into account when computing the distance (*Kaufman & Rousseeuw, 1990*).

## Beta diversity metrics

*Koren et al. (2013)*'s study was used as reference, because they evaluated beta diversity metrics' influence on the clustering of microbiome samples, which mirrors our goals. Their approach recommended comparing two or more distance measures in the clustering process, thus we chose five distinct distance measures from the list available in the R vegan package (version 2.3-1) (*Oksanen et al., 2015*). This was based on suggestions from the work of *Koren et al. (2013)* and a comprehensive study ranking all available beta diversity measures with abundance data (*Barwell, Isaac & Kunin, 2015*), that compared multiple quantitative and qualitative properties.

First, we selected well-extended Jensen-Shanon Distance (**JSD**), **rootJSD** and **Bray-Curtis**, as these are used in our reference study (*Koren et al., 2013*). In addition, because our method is independent of the availability of a phylogenetic tree associated to the OTU count matrix, we choose two additional metrics from *Barwell, Isaac & Kunin (2015)* to replace the phylogenetic tree-dependent Unifrac metric (weighted and unweighted) used by (*Koren et al., 2013*). Although there was a precedent study with 23 presence-absence beta diversity metrics (*Koleff, Gaston & Lennon, 2003*), we decided to focus on the richer metrics with continuous species abundance (*Barwell, Isaac & Kunin, 2015*). In general, abundance metrics are less biased than presence-absence ones when under-sampling. *Barwell, Isaac & Kunin (2015)* compares 29 beta diversity measures with 23 assorted properties. We chose the **Morisita-Horn** and **Kulczynski** metrics from the almost thirty analysed metrics, taking into account the overall ranking and some specific individual properties that are important for our use of beta diversity metrics as distance measures in clustering. Morisita is the highest scored beta diversity measure according to the comprehensive set of properties analysed in *Barwell, Isaac & Kunin (2015)*; although we must select the Morisita-Horn implementation (the third best scored) since our algorithm operates on normalized relative abundances. In addition, both Morisita and Horn-Morisita have been described as being "able to handle different sample sizes" (*Wolda, 1981*), which is an important characteristic in our exemplar studies, where the re-used microbiome datasets vary dramatically in size. Kulczynski, meanwhile, is the next-best ranked metric among those available in the R vegan

package, ranking sixth out of 29 metrics. Kulczynski is characterized as Pareto-dominant, and "found to have a robust linear (proportional) relationship until ecological distances became large" (*Faith, Minchin & Belbin, 1987*).

## Clustering assessment scores

*Koren et al. (2013)* recommend using at least two assessment clustering scores. The three complementary clustering scores we selected, and how they are included in our automatic procedure, are as follows:

1. **SI: average Silhouette width**: first, we search for the $k$ number of clusters with the best SI, with $k$ limited to the range of 2 to 10. This first step parallels the study of (*Gajer et al., 2012*), in which they also attempt to computationally define microbiome states. Here, we compute the average of all possible combinations of SI values for two different distance measures and each $k$, selecting the one with the highest average. The average SI must be greater than 0.25 in all selected measures, as this is the minimum threshold for "sensible" clusters according to *Rousseeuw (1987)*. This score takes into account the similarity between samples in the same and in the nearest clusters. If the selected pair of distance measures does not outperform the robustness constraint (see below), the next best combination is checked. SI was chosen as the means of selecting the best number of clusters because it is a standard, widely used metric to evaluate the quality of a grouping resulting from a clustering algorithm application; its utility is independent of the data source; and it can be used in absence of a gold standard as will be common in novel microbiome datasets.

2. **PS: Prediction Strength**: Although *Koren et al. (2013)* indicate that clustering selection could be restricted just to SI score for small datasets, we include an alternative, PS, which is also used in *Koren et al. (2013)*. Our method runs 100 repetitions where the dataset is split into two halves and clustering is applied on both. We then search for a correspondence between both groups of clusters, classifying a point in one half to the cluster in the other half, and vice-versa. Each pair is considered well-classified if both points classify to the same cluster in the other half. The score is the frequency of matching classification pairs. *Tibshirani & Walther (2005)* recommend that the selected number of clusters has a prediction strength score above 0.8.

3. **Jaccard similarity**: though the computation of PS implies some kind of bootstrapping (a resampling technique), our methodology allows an alternative, explicit step to verify the stability of the clusters selected based on the previous scores. This bootstrapping consists of a resampling with replacement, where clustering is computed over the whole dataset and, in addition, 100 resamples. Since the Jaccard score compares groups of elements, we compute the similarity of the original cluster with each resampling cluster. Thus, the resulting similarity score is the mean of the size of the intersection divided by the size of the union of samples. Following the guidelines for interpretation of Jaccard similarity (*Hennig, 2007*; *Hennig, 2008*), a stable clustering should yield a Jaccard similarity value of 0.75 or higher.

Clustering and the distances between OTU vectors were computed using a variety of R packages, including the *distance* function in the *phyloseq* R package (v1.19.1) (*McMurdie*

*& Holmes, 2013*), the *pam* function in the *cluster* R package (v2.0.6) (*Maechler et al., 2017*), the *hclust* function in the *stats* R package (v3.4.3) (*R Core Team, 2018*). Those algorithms take, as input, a distance matrix, where we use the metagenomics beta diversity measures comparing samples rather than a *{samples x features}* matrix as required by other algorithms. The first clustering assessment was computed with the *silhouette* function in the *cluster* R package; the robustness evaluation is computed with the *prediction.strength* function (*Tibshirani & Walther, 2005*) and the corresponding bootstrapping scored by Jaccard similarity with the *clusterboot* function (*Hennig, 2007*; *Hennig, 2008*), both in *fpc* R package (v2.1-11) (*Hennig, 2018*).

In summary, a total of 18,090 different clustering processes are executed to decide the best microbiome state definitions within a given dataset; 9 ($k = 2 : 10$) potential cluster-numbers × 5 distance measures (JSD, rJSD, Bray-Curtis, Morisita-Horn and Kulczynski) × 201 assessment scores (1 SI + 100 PS + 100 Jaccard) × 2 clustering algorithms (PAM and Hclust).

## Selected microbiome datasets

As stated in the introduction, our primary motivation in creating this algorithm was its application to longitudinal datasets, where we anticipate that the between-sample differences to be significantly lower than would be seen in, for example, population-based sampling studies. As such, these datasets represent particularly challenging examples with respect to identification of distinct states; moreover, there are few longitudinal datasets available in public repositories. We identified and selected eight such datasets, corresponding to different environment and ecological niches, and spanning a wide range of sample sizes and population complexities and distributions. Dataset details and availability are provided in the Data Citation section. Clearly, however, there is nothing precluding the use of this algorithm to inform any other microbiome study format.

The **human gut** microbiome dataset from *David et al. (2014)* is, to our knowledge, the longest and most frequently sampled longitudinal study of the human gut microbiome in healthy subjects. Briefly, it consists of near-daily stool sampling of two distinct subjects, throughout an entire year, including 493 gut samples with 4746 taxa. The input OTU table with absolute abundances was kindly shared by the authors in a personal communication. Samples from both donors are considered simultaneously in our study.

The **infant gut** microbiome dataset from *La Rosa et al. (2014)* has 922 samples and 29 OTUs at the class level. That collection includes data from 58 preterm babies, over different time points, for their first 1.5 months of life.

The **chick gut** microbiome dataset was generated by *Ballou et al. (2016)*. It analyses the response of the developing chick gut microbiome to different treatments (salmonella vaccine and/or probiotics) during their first month of life. The dataset consists of 119 samples with 1583 taxa. The samples include six time points, with 4 or 5 subjects per each of the four treatment combinations.

The **lake** microbiome dataset *Dam et al. (2016)* come from the freshwater Lake Mendota in USA, with 91 samples over 11 years and 17462 taxa, where approximately 5% of the taxa constitute 80–99% of the total microbiome.
The *Gajer et al. (2012)* **vaginal** microbiome dataset consists of 937 samples and 330 OTUs, corresponding to 32 women, with samples collected twice per week for 16 weeks. Dataset details and availability are provided in the Data Citation section. In this case, the original data counts are already pre-processed, and normalized to a sum of 100 per sample, as relative abundances. This contrasts with the previous datasets, where a normalization procedure was applied by us before entering the data into our algorithm.

The **left and right palm and tongue** microbiome datasets are longitudinal datsets from the Human Microbiome Project, analysed in *Caporaso et al. (2011)*. That study collected samples from two individuals (man and woman) at four body sites at up to 396 timepoints. In the case of tongue, we selected only the samples from donor F4 as the other samples were not amenable to normalization following the (*David et al., 2014*) methodology, which was used for the other samples.

## RESULTS

### Microbiome state determination including all taxa

We applied our state-identification algorithm to each of the eight selected dataset using each of the three complementary clustering assessment steps defined in the Methods sub-section "*Automating the identification of granular, yet robust states*". A representative result is provided in the eight panels of Fig. 1, where PAM was selected as the clustering approach, and corresponding to the eight independent datasets. Outputs using HCLUST are available in the Fig. S1. Each graph includes the scores of five distinct diversity metrics, corresponding to the colored lines. The three graphs within each panel of the figure represent the following analyses:

1. The first graph shows the results of the algorithm's attempt to choose the most suitable number of clusters, $k$, according to distinct beta diversity measures (i.e., distance among samples) scored according to SI. The selected $k$ value (from 2 to 10) must report the highest average SI in the best pair of two beta diversity measures. For example, in Fig. 1A, the number of clusters selected would be $k = 3$, since that is where the algorithm detects an SI score with the highest value (0.602; utilizing the PAM algorithm, and the JSD metric). The other metrics are also higher than the minimum threshold, with Morisita-Horn being the second best metric, scoring above the threshold for strong clusters.

2. The second graph shows the test of whether the $k$ value chosen by SI in the first graph is sufficiently robust, by testing it against the PS criterion of being greater than 0.8. The second graph of Fig. 1A shows that JSD and rJSD in PAM with $k = 3$ satisfies the robustness test, providing PS scores of 0.950 and 0.935 respectively. At $k = 4$, however, the PS value decreases to below the acceptable threshold for all metrics, thus showing that, despite $k = 4$ being an acceptable number of clusters based on the SI, classification of the data into four clusters is not robust.

3. Finally, the third column graph shows the stability of the selected $k$ clusters, by testing if the Jaccard similarity of the chosen diversity measures exceed 0.75. The third graph of Fig. 1A verifies the stability criterion, with Jaccard $= 0.986$ for the previously suggested $k = 3$ clusters for JSD and, in this case, also for all remaining beta diversity metrics.
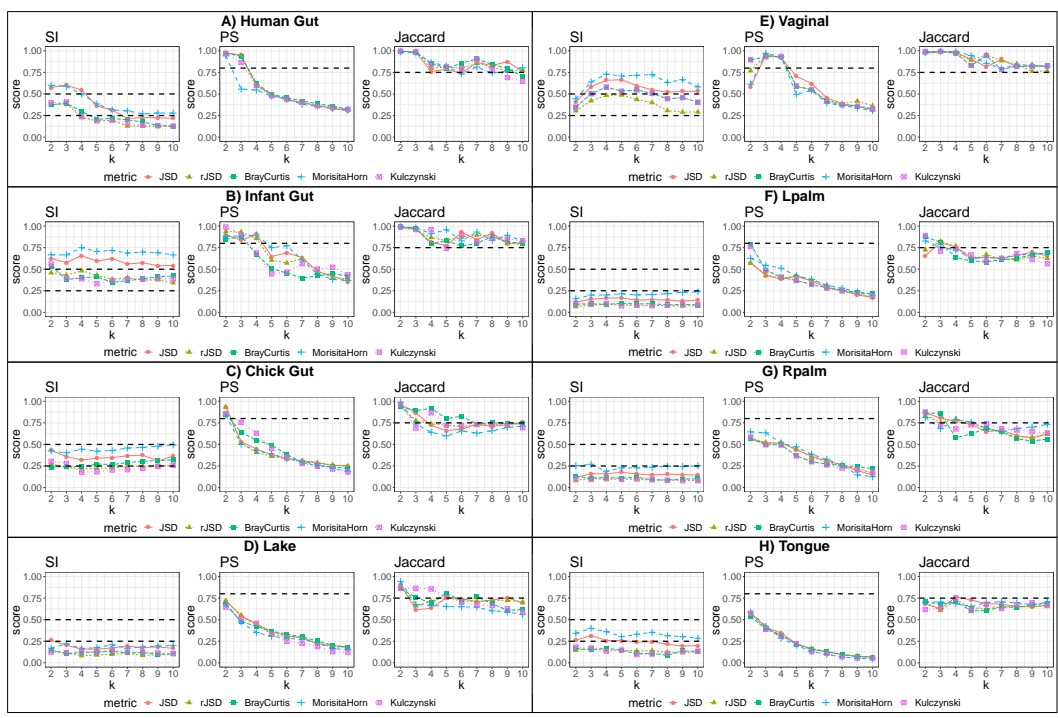

**Figure 1  Robust clustering evaluation using the PAM algorithm on eight datasets including all taxa.**
(A) Human gut. (B) Infant gut. (C) Chick gut. (D) Lake. (E) Vaginal. (F) Lpalm. (G) Rpalm (H) Tongue.

For the remaining datasets, Figs. 1B, 1C and 1E , i.e., infant gut, chick gut and vaginal microbiome, show similar patterns to the human gut dataset: the various assessment measures, for some number of clusters, exceed the established thresholds (dashed horizontal lines), suggesting that the data can reliably be classified into that number of distinct states. Conversely, Fig. 1H shows that the tongue microbiome fulfills the SI evaluation for well-defined clusters; however, the PS and Jaccard analyses demonstrate that none of these clusters are sufficiently robust to represent a genuine state. Finally, for the lake microbiome, and the left and right palm microbiome (Fig. 1D–1G), our pipeline determines there are no clusters whatsoever, when taking all taxa into account.

### *Association of states with biological phenomenon*

Beyond the statistical support provided by the robustness measures above, we undertook to determine if the state-identification and classification algorithm was able to detect states that were associated with observed biological phenomenon. The three states we identified in the human gut dataset from *David et al. (2014)*, were tightly associated with their sample annotations: {*subject A, subject B* before infection, *subject B after infection*} (see Fig. S2). For subject A, there was a period of travel; however, their gut microbial composition was indistinguishable before and after the travel. In subject B, two states were detected, corresponding to the samples before and after a short period during which the subject experienced an infection.

In the chick gut microbiome, the two clusters identified by our method largely correspond to *immature/young* chicks and *mature/adult* chicks, reinforcing the conclusion of the original manuscript, which showed chick age to be the primary differential factor between samples, as it is described in the subsection titled "Exemplar application - state sequence diagram".

In the vaginal microbiome dataset, our algorithm determined that the strongest supported grouping consisted of four states (clusters), rather than the five clusters identified in the original manuscript (*Gajer et al., 2012*). These four states were clearly assignable to the clusters identified in their manuscript as I, II, III, and IV-B (see Table S1). Members of their cluster labeled IV-A are distributed among the other states identified by our algorithm, specifically in cluster I (our state 4) and cluster III (our state 1), and several with cluster IV-B (our state 3). We note that, based on our algorithm, Prediction Strength decreases dramatically when the number of clusters increases from $k = 4$ to $k = 5$. According to the original authors, only cluster IV-B is associated with a notable disease state (bacterial vaginosis), while the remaining clusters correspond to a (nominally) healthy vaginal microbiome with no phenotypic distinction between the subjects in those clusters (*Gajer et al., 2012*). Thus there is no biological evidence arising from this study contradicting our assignment of only four robust states in this dataset, versus the five clusters described in the original study. The results of the vaginal microbiome are explored in greater detail in the subsection titled "Relationship to other definitions of microbiome" below.

Without any published data providing an expert classification of the preterm infant gut microbiome dataset from *La Rosa et al. (2014)*, our robust clustering methodology determined that the optimal number of states that could be robustly partitioned was $k = 4$. When we analyse the microbial composition of these 4 states, we found that state 2 contains ~50% of the samples, state 1 contains ~25%, state 3 ~20% and state 4 has <5% of the samples. State 3 appears to include the set of youngest babies; State 1 the oldest babies; and state 2 groups those of intermediate age. Analysing the microbial composition of samples in each cluster, we found state 1 and 3 are mainly enriched in Firmicutes, state 2 in Proteobacteria and state 4 in Bacteroidetes (see Fig. S3), a distribution that aligns with the results reported by *La Rosa et al. (2014)* where they suggest a beginning-of-life with primarily Bacilli (phylum: Firmicutes), followed by a period with Gammaproteobacteria prevalence (phylum: Proteobacteria), and finally, a gut with Clostridia as the dominant species (phylum: Firmicutes). We could not determine a biological association for the small cluster 4, based on the metadata provided by (*La Rosa et al., 2014*).

The remaining microbiome datasets we selected did not reveal clear separation into states, likely indicating that the microbial composition over time within the samples in those datasets did not exhibit significant variations. This may be due to the lack of external perturbations between sampling periods and/or of including samples belonging to only one (lake and tongue) or two (left and right palm) subjects, reducing variability of the samples, and thus the possibility of the data including distinct microbiome states.

**Table 1  Number of clusters resulting from analyses using different taxa subsets, and data aggregations.** D/ND, Dominant/Non-dominant taxa subset at 1%. A dash indicates that no robust clusters were identified by our algorithm. The "default taxonomic level" is, for most datasets, Species (for the vaginal microbiome it is Class-level).

| Dataset | Taxa subset | | | | | |
|---|---|---|---|---|---|---|
| | Default tax. level | | | Genus | | |
| | All | D | ND | All | D | ND |
| Human gut | 3 | 2 | 3 | 2 | 2 | 2 |
| Infant gut | 4 | 3 | 4 | – | – | – |
| Chick gut | 2 | 3 | 6 | 2 | 2 | 4 |
| Lake | – | 2 | – | – | 2 | – |
| Vaginal | 4 | 5 | – | – | – | – |
| LPalm | – | 2 | – | – | 2 | – |
| RPalm | – | 4 | – | 2 | 2 | – |
| Tongue | – | – | – | – | – | 2 |

**Table 2  Number of taxa (i.e., OTUs) using different data sub-settings and aggregations.** D/ND, Dominant/Non-dominant taxa subset at 1%. A dash indicates that taxonomic aggregation was not possible, due to not enough taxonomic data.

| Dataset | Taxa subset | | | | | |
|---|---|---|---|---|---|---|
| | Default tax. level | | | Genus | | |
| | All | D | ND | All | D | ND |
| Human gut | 4,746 | 19 | 4,727 | 386 | 19 | 367 |
| Infant gut | 29 | 3 | 26 | – | – | – |
| Chick gut | 1,583 | 6 | 1,577 | 180 | 9 | 171 |
| Lake | 17,462 | 18 | 17,444 | 1,607 | 26 | 1,581 |
| Vaginal | 298 | 13 | 285 | – | – | – |
| LPalm | 15,214 | 16 | 15,198 | 1,378 | 17 | 1,361 |
| RPalm | 17,676 | 14 | 17,662 | 1,535 | 17 | 1,518 |
| Tongue | 1,631 | 15 | 1,616 | 329 | 11 | 318 |

## Determining the source of the robust state calls

In this section we examine if and how state identifications change when we execute the algorithm using only subsets of the input data, based on inclusion or exclusion of various taxa. The results are shown in Table 1. First, we select only the dominant taxa (top 1%: D) and the inverse set (bottom 99%: ND) as input to the algorithm (number of taxa available in Table 2). This was done to determine if the identification of state-clusters was primarily reliant on the most dominant taxa, or if a significant "signal" was emerging from the remaining taxa. Second, where the source data allows, we aggregate taxa at a higher resolution (at Genus taxonomic level) and re-execute the analysis of all, dominant and non-dominant taxa at the genus (see columns "Genus" in Table 1). This was done because, in many studies, OTU tables are aggregated at the Genus level. As such, we wished to demonstrate that the algorithm can perform on data organized in this way.

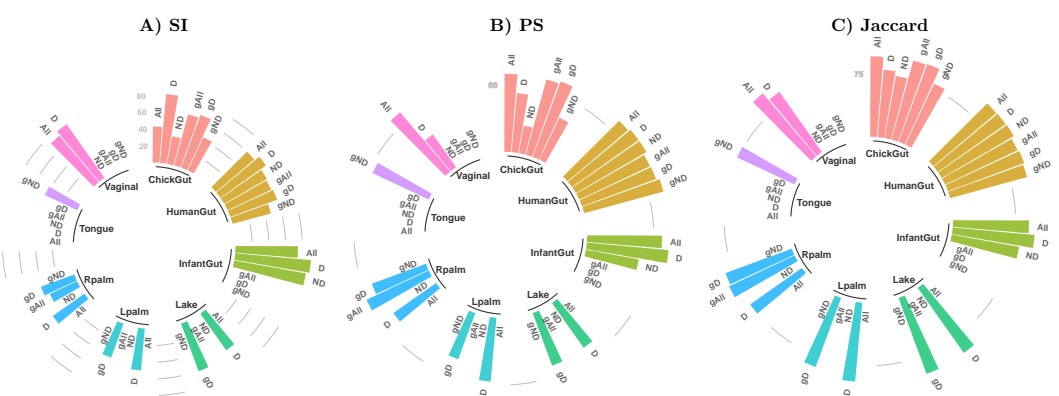

**Figure 2 Assessment of robust clustering results.** (A) SI, (B) PS, (C) Jaccard. For all, dominant (D) and non-dominant (ND) taxa subset at 1%, at default and Genus-aggregated taxonomic levels (gAll, gD and gND). No bar is shown when no robust clusters were identified. Clustering quality scores are represented with grey lines in percentage (SI >25% and 50% and (PS >80% or Jaccard >75%)).

Table 1 and Fig. 2 show a summary of the results of running our algorithm on data with these various groupings and taxonomic subsets, and Table 2 shows the data distribution using these filters. The columns labelled "All" in Table 1 correspond to the results described in detail in the previous sections, using all taxa for comparison. The first notable observation is that, using only dominant taxa at the default taxonomic level (second column), our algorithm identified clusters in almost all datasets, whereas using only non-dominant taxa, very few robust states were identified. On the other hand, aggregating OTUs at the Genus level generally reduces the number of robust states identified. These results suggest that the clustering "signal" is coming largely from the species level, and primarily from the dominant taxa in the sample. The exception to this was the chick gut, which reveals 6 robust states (the highest value observed in our study) when using only non-dominant taxa at the species level. Interestingly, in this case, the inclusion of dominant taxa masks what appear to be genuine, biologically-meaningful states, in that these six clusters align with the increasing age of the chicks (see Fig. 6 from *García-Jiménez, De la Rosa & Wilkinson, 2018*).

We then examined the clustering assessment measures on these selected and aggregated data, with the results represented in Fig. 2. The best Silhouette (SI) values (the highest bars) are achieved in the infant gut microbiome for all three taxa subsets (all, dominant, and non-dominant), followed by vaginal, chick and human gut datasets. The palm datasets, tongue, and lake microbiomes have the lowest SI values, indicating a less distinct split of these samples into states, if states were identified at all. Nevertheless, the presence of a bar in the plot indicates that the quality threshold was met. According to the Prediction Strength (PS) measure, the distribution of values is similar to that of the SI scores, although with some variations in several taxa subsets within the same dataset. Finally, in terms of clustering stability as analysed by the Jaccard similarity measure, the values for all datasets are high, and similar to one another.

Figure 3 shows the differences in the distribution of samples into state-clusters using the six filtered/aggregated data subsets of the gut microbiome dataset (see Fig. S4 for
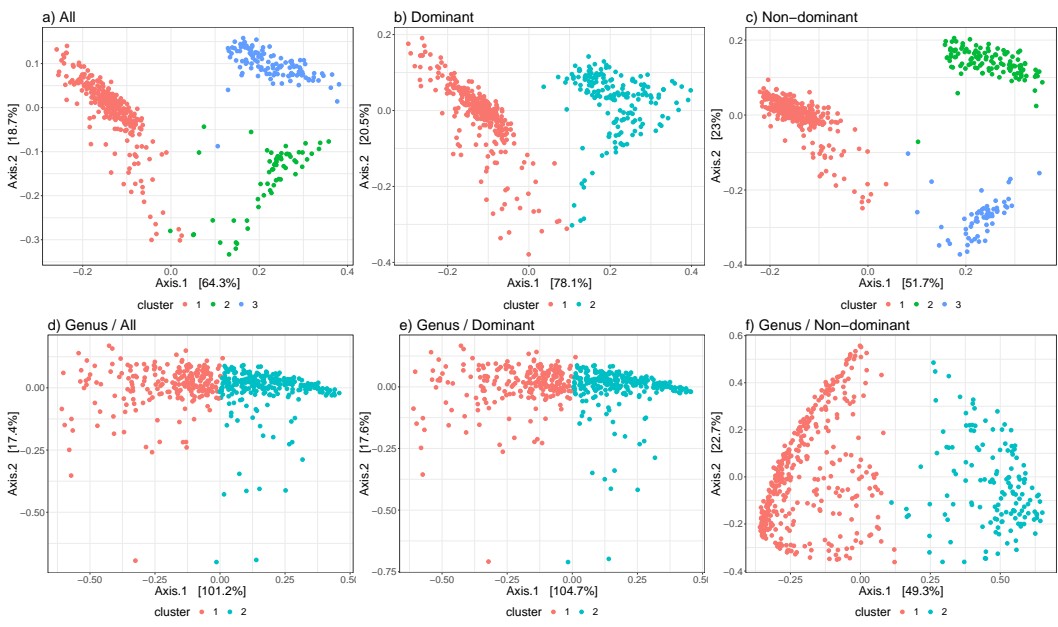

**Figure 3** **Microbiome clusters from Human Gut with varying filters and aggregations, represented as Principal Coordinates graphs.** (A–C) Species level data. (D–F) Genus-level aggregation.

corresponding results in the chick gut dataset). In Figs. 3D and 3E (Genus aggregation, all and dominant taxa, respectively) clusters are difficult to distinguish, suggesting again that species-level information contributes to the quality of cluster separation. The point distribution in the scatterplot also depends on the beta diversity metric, with the best metric being JSD for data at species-level (shown in the top row) and the Morisita-Horn metric for data at Genus-level (shown in the bottom row).

Finally, Table 2 shows convincingly that our algorithm can operate successfully over a wide range of data profiles—from a few (<10) taxa, to many thousands; moreover, that it can operate at the Genus or the Species level (though Species is demonstrated to be more informative).

## Relationship to other definitions of microbiome "state"

Other studies have explored microbiome states using a variety of clustering algorithms, population granularity, and data pre-processing pipelines. These studies then use a variety of terms to describe the clusters of similar microbiomes that emerge from their analyses. For example, with population-level human gut microbiome studies, the term "enterotype" emerged to describe the shared set of microbial community features between large segments of the population. Other studies, looking at other body cavities, have coined phrases such as "state type" or "community groups" to describe a clustering of microbiome community structures, at a given level of granularity. The selected level of granularity, however, is somewhat arbitrarily decided, and indeed, several authors recognize this by indicating that they observe what appear to be sub-groups within their defined groupings.
To demonstrate how our concept of "state" fits into this milieu of varying descriptors, we reanalyse the data of *Ravel et al. (2011)*. The vaginal microbiome dataset from (*Ravel et al., 2011*) yielded, by their analysis, five "community groups". Each group was associated to its probability of the incidence of bacterial vaginosis, and also the proportion of individuals with different ethnicities within that group.

We applied our algorithm, independently, to members of each of the more populated community groups from that study: I, III and IV. This analysis yielded 2, 3 and 2 additional, more fine-grained states (respectively) as the highest-scoring states within these community groups (see Fig. 4). The data potentially support an even larger number of more fine-grained states, given that higher cluster numbers are also supported by robust SI and Jaccard scores. These finer-grained states may relate to some detectable biological reality—for example, the two states identified within in community group IV segregated almost all women of the black ethnicity (excluding three) and many of the Hispanic group, into the same state. A researcher could thus test the various sets of statistically-supported states emerging from our algorithm to determine which, if any, correlate to their biological feature(s) of interest, then use these state definitions as biomarkers for their study. This also shows that the same data can be interrogated, iteratively and without manual adaptation of the algorithm, to reveal statistically supported novel substructures that could act as higher-granularity biomarkers applicable within that subset of individuals.

We also reanalyze the Human Gut microbiome *David et al. (2014)* datasets, now, separately for subject A and B, to compare with the conclusions of the original study. Regarding subject A, *David et al. (2014)* defined two states, corresponding to before/after and during traveling periods (before and after travel, the state is the same). Thus for subject A we get two clusters, which is in agreement with Figs. 3B and 3D of the original study (*David et al., 2014*), where one cluster represents samples taken during travel (approx. 70–120 collection days) and other cluster containing the remaining samples (see Fig. S5). On the other hand, with respect to subject B, Figs. 3F (and 3H) from the original study (*David et al., 2014*) identifies three clusters. Our pipeline finds only two clusters, one with samples before infection and the other after. The infection period is represented by very few samples (approximately five samples), and using our statistical cutoff, this small number of samples is insufficient to be considered a robust cluster; however, in our study of filtered data, using the subset of dominant taxa aggregated by genus (four clusters, SI $= 0.5$, PS $=$ $0.49$, JC $= 0.84$), a new cluster appears containing the samples taken during the infection period, although in these results the pre-infection period is now divided in two clusters (see Fig. S6). Thus, our observations largely align with the observations of *David et al. (2014)*, with slight differences emerging when we change the granularity of the input data.

That we obtain highly comparable, though non-identical results to previously published studies, using the same data, is not unexpected. Many choices are made in the course of a microbiome study, and as we point-out in our introduction, the bases for these choices are not widely standardized or shared within the community. Making different choices, e.g., for the approach to clustering, or the statistical bases for declaring a cluster as "valid" will clearly affect the final outcome of the study. Our algorithm objectively identifies clusters that can be rigorously statistically supported as to their robustness. In those cases where

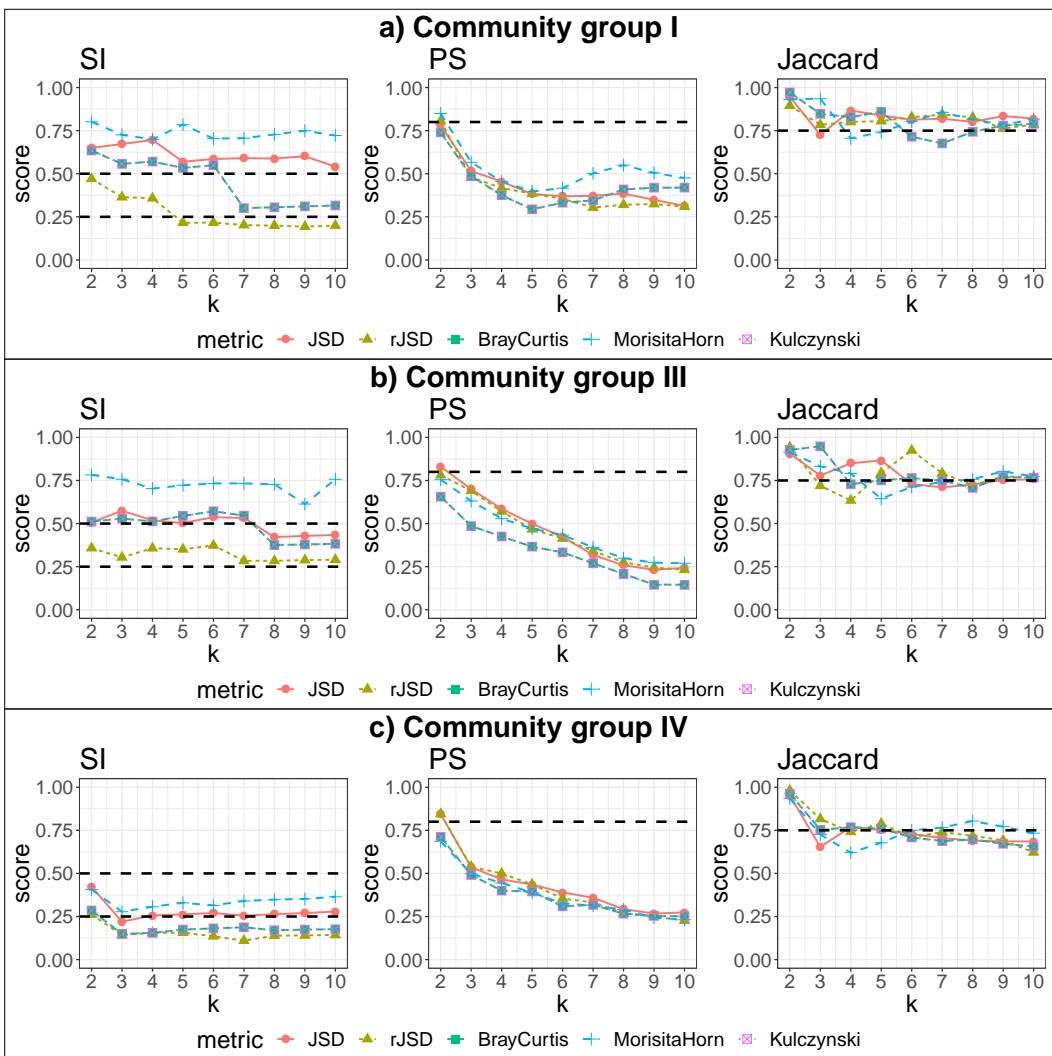

**Figure 4** **Robust clustering evaluation, finding states within community groups.** Using the PAM algorithm on three groups defined in *Ravel et al. (2011)* including all taxa. (A) Community group I, (B) Community group III, (C) Community group IV.

we found fewer (or more) clusters than described in the published studies, we were able to use these statistics to show that there is a more robust interpretation of the data than that previously published.

Overall, then, the algorithm we have defined is applicable to data at many levels of granularity, and finds robust "states" within those data. It self-adjusts to the level of dataset and sample complexity, and provides the set of states that are statistically rigorous within that dataset.

## Comparison to related approaches

This section compares our automatic and data-driven definition of states in longitudinal microbiome datasets to related techniques. There are tools focused on clustering

(independently of microbiome applications), others on time series in general, and yet others specifically on the analysis of microbiome datasets. Among those, we have selected those approaches that provide the most sensible comparators, although they generally have distinct goals from ours. We describe their similarities and differences in Table 3.

Table 3 shows that the tools and approaches that could appropriately be applied to analyse microbiome clustering and microbiome dynamics over time are very diverse, differing by the inputs, outputs, assumptions, and primary objectives of the approach. Thus, it is difficult to identify valid metrics that would enable a sensible quantitative comparison between them.

The clustering techniques used in this study are not novel, but the methodology of their application to longitudinal microbiome time series to identify states is distinct from other approaches. The main differences of our approach are that we group samples—taking samples from more than one subject into account—and we allow non-sequential microbiome samples to appear in the same cluster; in addition, we allow microbiomes states to be repeated at various points within the same or different subjects. We will now provide specific comparisons with other approaches, highlighting the distinct methodological and/or statistical differences in our approach.

Ananke (Hall et al., 2017) is applied to temporal microbiome data to analyse its dynamics. The Ananke analysis is based on a clustering of sequences, in contrast to our approach of clustering samples. Ananke takes, as input, FASTA sequences, while our approach consumes an abundance matrix, i.e., a samples × OTU matrix, with the OTUs predefined. Thus, the elements we group in our clustering are microbiome samples represented as a vector of abundances of different OTUs, with the same OTUs potentially appearing in different samples. As a result, our approach is able to group samples from more than one time series - i.e., from several subjects - versus the single time-series that is the input to Ananke. This then highlights our distinct goal, which is to discover comparable microbiome states between different individuals, and even if they appear in non-sequential order. An additional distinction is that our approach attempts and evaluates a wide spectrum of different parameter-values in its search for the optimum clusters, while Ananke has coded defaults and/or user-defined parameters (which may be incorrectly assigned by the user). This makes our approach both more objective, as well as amenable to automated environments, such as that used in the MDPbiome (García-Jiménez, De la Rosa & Wilkinson, 2018) study, without modifying the default behavior of the algorithm.

TIME (Baksi, Kuntal & Mande, 2018) is a microbiome time series visualization tool, supporting biologists' attempt to gain insights from microbiome taking time into account. Among its available analyses, TIME allows the user to cluster taxa or time points; however, it is not able to group microbiome samples in a global way, over multiple subjects, as our approach does. Moreover, TIME is specific to longitudinal analyses, where our algorithm is applicable more broadly.

NetShift (Kuntal et al., 2019) explores differences between exactly two pre-defined states (e.g., health vs disease), based on differences in their microbe interactions. It takes as input two association networks, and compares their global and local graph properties (density, average path length, degree, etc.). A key difference, therefore, is that we take an objective,

**Table 3  Comparison of methodological approaches applicable to microbiome clustering and dynamics analysis.**

| | This study | Ananke (*Hall et al., 2017*) | TIME (*Baksi, Kuntal & Mande, 2018*) | NetShift (*Kuntal et al., 2019*) | MDSINE (*Bucci et al., 2016*) | MDPbiome (*García-Jiménez, De la Rosa & Wilkinson, 2018*) |
|---|---|---|---|---|---|---|
| Input | OTU table | sequences, time points | OTU table, metadata | 2 association networks | OTU table, qPCR | microbiome states, perturbations |
| Output | microbiome states (samples groups) | sequence groups (TSC) | visualizations over time | network scores visualizations | concentration, interaction prediction | state transition and perturbation prediction |
| Clustering | group samples | group sequences | group taxa/time points | no | group taxa | no |
| Suited to longitudinal/static | yes/yes | yes/no | yes/no | no/yes | yes/no | yes/no |
| Suited to > 1 subject | yes | no | no | yes | yes | yes |
| Suited to > 2 states | yes | yes | yes | no | yes | yes |
| Predict interactions | no | no | yes | yes | yes | no |
| Suited to perturbations | no | no | no | no | yes | yes |
| Automatic pipeline/ User-Interaction interpretation | A | UI | UI | A/UI | A | A |

data-centric approach to computing microbiome states, which is independent of any external biological observation; moreover, we allow multiple states, as well as a temporal sequence of states, compared to the two pre-defined static states that are input to NetShift.

MDSINE (*Bucci et al., 2016*) analyses microbiome time series data in terms of interaction networks, suggesting if/how yes-no perturbations affect the microbial composition. MDSINE does not define groups of microbiome samples, but rather taxa sub-communities. The key distinction, therefore, is the goal of MDSINE to predict taxa concentration, which is distinct from the definition of a statistically-defined "state" that can behave as a biomarker. Further, MDSINE requires qPCR data as input, in addition to the OTU table that is unique input to our algorithm.

MDPbiome (*García-Jiménez, De la Rosa & Wilkinson, 2018*)—the motivating study that led to the development of this algorithm - is capable of consuming a set of microbiome states as its input (such as those identified by this algorithm), and then undertakes a further analysis related to the state-transitions that appear within each individual in the dataset, and the correspondence between those state transitions and known external perturbations.

### Exemplar application—state sequence diagram

Taking into account our interest in the interpretation of longitudinal microbiome datasets, we have applied our algorithm to time-series datasets, keeping the time-sequence information as a vector. This generates a diagram showing the sequence of microbiome states that individual study subjects follow along the time-axis. Such diagram could help with biological interpretations related to, for example, the sequence of state changes, the temporal correlation between state-changes and external perturbations, and the set of possible state-to-state transitions. Figure 5 shows examples of these diagrams for chick gut and vaginal microbiome datasets. Other exemplars are available in our Zenodo deposit (see "Data availability" section).

These diagrams are consistent with the observations made by the original authors of these datasets, showing that, for example, that the chick gut microbiome follows a linear path from adolescence to maturity, and that there are stable 'ground states' in the vaginal microbiome that differ from individual-to-individual, and that persist even after brief perturbations (*Ballou et al., 2016*). Thus, our algorithm identifies microbiome states consistent with previously published studies, but does so automatically, objectively, and with rigorous statistical support.

## DISCUSSION

There are a variety of choices that must be considered when clustering metagenomics data, and in many existing approaches, these decisions may be made subjectively, and/or are not subsequently validated. For example, testing whether clustered data is significant and robust, or selecting clusters based on non-statistical information such as phenotype, where the data interpretation and manipulation is driven by the presumed outcome. After reviewing and analysing many studies that followed distinct approaches to clustering gut microbiome datasets, *Costea et al. (2018)* concludes that no procedure is obviously preferred over any other, but rather, that the selection of an approach will depend on

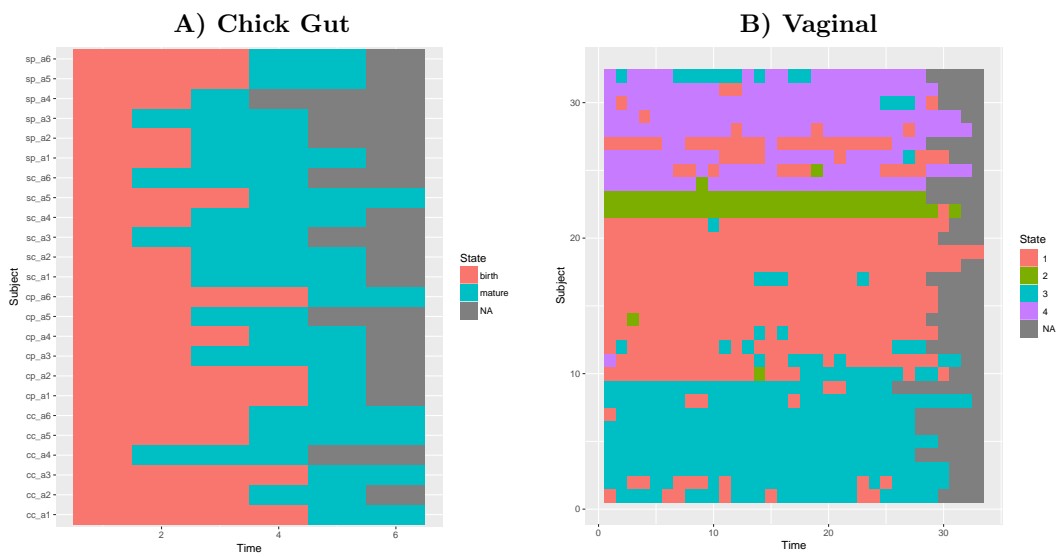

**Figure 5** **Cluster time series diagrams.** (A) Chick gut, (B) Vaginal. Each row represents a different subject, and each column represents a time point in the longitudinal sequence. Grey points represents no additional time point for that subject.

the experimental conditions and/or the features of the resulting data in each particular case. While this observation is true, it is somewhat unsatisfying since, in the absence of an objective standard that arrives at a statistically-supportable conclusion, it is difficult to do important comparative studies or to reanalyse the data using different features of interest or domain expert opinions.

Here, we respond to the need for standardization in microbiome analyses, promoted by the International Human Microbiome Standards (IHMS) (*Morton et al., 2017*) and the Microbiome Quality Control Project (MBQC) (*Costea et al., 2017*) with a reproducible, flexible, objective, internally-validating pipeline to define states within microbial populations, thus enabling comparability. In this approach, a wide array of distinct clustering procedures are automatically executed, and the best one for the given dataset is highlighted, where the designation of "best" is supported by clustering assessment measures. The set of procedures tested within the pipeline include approaches that would be appropriate over a wide data-space, including highly complex population-level studies, to lower-complexity (e.g., intra-individual) studies, and with data captured at a variety of taxonomic levels. This data-driven approach reduces the subjectivity of metagenomics analyses, providing the most statistically-supportable outcome, regardless of the input data, and including null outcomes.

Our approach requires certain assumptions. In particular, we presume that even data that varies continuously can be modeled in a discrete manner—that the continual flux in community makeup can nevertheless be resolved to particular states that will appear as statistically significant groupings. While there is disagreement in the microbiome community about whether this assumption mirrors any reality in the microbial population dynamics, our discretization approach is, we believe, a plausible simplification of
microbiome dynamics. In fact, many real processes (including many biological ones) are not discrete, although they are simplified as discrete to allow their modeling to be studied with computational and mathematical models (*Faust & Raes, 2012*; *Faust et al., 2015*), as we do here by defining microbiome states within a given dataset.

The clustering outcomes on the discretized data are then tested and validated using a variety of metrics. PS seems to be the most restrictive score that can be applied to discriminate between a viable partition of microbiome samples, and other non-robust possibilities where the scores fall below its threshold of 0.8 (see central columns of Fig. 1). Regarding beta diversity measures, JSD and Morisita-Horn are the metrics that usually resulted in the highest SI values, and this held true even for the dominant/non-dominant filtered data subset, and the aggregated taxa subset.

With respect to the utility of the identified states, the approach attempts to maximize the number of statistically supportable clusters, with the aim that the resulting state definitions will have a sufficient granularity to be useful as biomarkers. In this regard, we attempt to avoid singletons or very small clusters (with size $< 5$), and note that, in this study, Hclust tends to often generate just two clusters according to the limit established by the robustness PS score (see central columns of Fig. S1). The final bootstrapping step to assess cluster stability, measured in Jaccard similarity terms, determines that PAM partitions represent valid and stable clusters, with almost all $k$ values. This contrasts with Hclust, where many $k$ values with different distance measures do not reach the minimum of 0.75 of mean Jaccard similarity. As such, we suggest that Hclust is perhaps not suitable for this task, while an agglomerative approach (e.g., PAM) is more suitable than hierarchical partitioning, given our desired objectives and results. Within the eight datasets used in this manuscript, our approach identified as many as six different microbiome states that passed robustness challenges. There is no "gold standard" that can be used to assess these results; nevertheless, the states identified seemed to correlate, in most cases, with the biological knowledge about the samples that could be gleaned from the available sample metadata, suggesting that they represent, as intended, genuine biomarkers of biological relevance.

Beyond observations about the algorithm itself, the various result data from our study provide observations that are potentially of-interest. First, the relative importance of dominant and non-dominant taxa with respect to cluster number and robustness is non-trivial. Table 1 shows that removing the top 1% dominant taxa ("ND" column) sometimes reduces the overall ability of the algorithm to identify clusters; however, in other cases, additional clusters that pass robustness tests, and seem to correlate with biological annotations, are revealed, indicating that in these cases minor population changes are being masked by the dominant species. Previous studies found that enterotypes could largely be defined based on the differing proportions of a small number of dominant species (*Gorvitovskaia, Holmes & Huse, 2016*), and this is largely consistent with our observations. Nevertheless, the inclusion of non-dominant species ("All" column) affected the clustering results in all cases, compared to the use of dominant taxa alone. Focusing on less abundant taxa to characterize microbiome clusters is not common practice, though some studies highlight interesting events when doing so. For example, *Claussen et al. (2017)* found there are interactions between low abundance taxa, and *Martí et al. (2017)* concludes that the

most abundant taxa are less volatile than the less abundant ones. Our results similarly suggest that giving attention to dominant and non-dominant taxa, independently, may sometimes yield insights. This is particularly true in cases such as our primary use-case where we follow the dynamics of the microbiome within individual subjects, thus do not necessarily anticipate dramatic shifts in dominant species. Our algorithm is agnostic to the selection of input taxa, and thus users may decide to do these additional studies if it is suitable for their goals.

As with most machine-learning approaches, this approach is sensitive to the sample size, failing when there are too few samples to represent the number of true states, or when particular states are exceedingly rare. In such cases, and depending on the final objective, our results suggest that focusing on dominant taxa may assist in discovering robust states, even if some states cannot be discriminated.

Finally, we return to the issue of relevance. Our objectives in this study were twofold. First, to generate an approach and algorithm for detecting the state of a microbial community that would be broadly applicable to a wide range of microbiome study designs, and that would therefore better-harmonize the analytical approaches between studies, as well as add rigor to studies which currently do not statistically validate their compartmentalization of the various microbial community structures. Second, we wished to design an algorithm that could objectively define microbial community states in the absence of any gold-standard, or even any phenotypic expression of that state, such that these states could be used as a microbiome-based biomarker (*Gorvitovskaia, Holmes & Huse, 2016*). We demonstrate here that we can identify microbiome states that, on comparison, align with expert biological annotations when available, but also that the states we identify correlate with perturbations that are known to affect microbial communities - for example, changes in diet, maturation, and disease. Thus, we believe that these states do, in fact, achieve this second goal. We have utilized this in a recently published study (*García-Jiménez, De la Rosa & Wilkinson, 2018*) where we use these biomarkers to track the evolution of a variety of microbiomes as they respond to, and/or recover from, external perturbations.

## CONCLUSIONS

This manuscript describes an automated algorithm that determines a set of distinct microbiome states, given a set of related samples—multiple samples from within a particular niche, or multiple samples from the same niche over time. Our novel methodology is characterized by a robust, objective, transparent and reproducible assessment of the quality of the identified states. The algorithm is flexible with regards to the data source, pre-filtering, and taxonomic level of the samples, and may be applied to a wide range of investigations over diverse species and experimental designs. We make all outcomes from this work publicly available, with the desire that it will lead to a more harmonized, consistent, rigorous, and comparable studies going forward.

### Data Citation

This section describes the sources of the different input datasets.

***Human gut microbiome (David et al., 2014):***
- Metadata: David, L. et al. *Genome Biology*. Additional file 18: https://static-content.springer.com/esm/art%3A10.1186%2Fgb-2014-15-7-r89/MediaObjects/13059_2013_3286_MOESM18_ESM.csv
- OTU table: it was kindly shared by the authors in a personal communication
- Raw data: *EBI/ENA* ERP006059 (2014)

***Chick gut microbiome (Ballou et al., 2016):***
*Qiita* (*Gonzalez et al., 2018*) ID 10291.

***Infant gut microbiome (La Rosa et al., 2014):***
OTU table and metadata: La Rosa, P.S. et al. *PNAS* Supporting Information, Dataset_S01: http://www.pnas.org/lookup/suppl/doi:10.1073/pnas.1409497111/-/DCSupplemental/pnas.1409497111.sd01.xlsx

***Lake microbiome (Dam et al., 2016):***
*Qiita* (*Gonzalez et al., 2018*) ID 1242 and www.lter.limnology.wisc.edu.

***Vaginal microbiome (Gajer et al., 2012):***
- OTU table and metadata: Gajer P. et al. *Science* Table S2 (2012)
- Raw data: *SRA* SRA026073 (2012).

***Left and right, tongue microbiome (HMP) (Caporaso et al., 2011):***
*Qiita* (*Gonzalez et al., 2018*) ID 550.

***Vaginal microbiome—Community groups (Ravel et al., 2011):***
OTU table and metadata: Ravel et al. *PNAS* Table S4 (2011)

## ACKNOWLEDGEMENTS

Thanks to the authors of *David et al. (2014)* for kindly providing the OTU table privately and to the authors of *Ballou et al. (2016)* for pleasantly answering our questions about their datasets and metadata. All authors had full access to all the data in the study and take responsibility for the integrity of the data and the accuracy of the data analysis. We thank Dr. Keith Walley for his inspiration in convincing us of the need for this algorithm.

### Funding

Mark D. Wilkinson is funded by the Ministerio de Economía y Competitividad grant number TIN2014-55993-RM, and by the Isaac Peral programme of UPM. Beatriz García-Jiménez is funded through an award from the Severo Ochoa programme of the CBGP UPM-INIA Severo Ochoa Center of Excellence, Madrid. The funders had no role in study design, data collection and analysis, decision to publish, or preparation of the manuscript.

## Grant Disclosures

The following grant information was disclosed by the authors:
Ministerio de Economía y Competitividad: TIN2014-55993-RM.
Isaac Peral programme of UPM.
CBGP UPM-INIA Severo Ochoa Center of Excellence, Madrid.

## Competing Interests

The authors declare there are no competing interests.

## Author Contributions

- Beatriz García-Jiménez conceived and designed the experiments, performed the experiments, analyzed the data, contributed reagents/materials/analysis tools, prepared figures and/or tables, authored or reviewed drafts of the paper, approved the final draft.
- Mark D. Wilkinson conceived and designed the experiments, analyzed the data, authored or reviewed drafts of the paper, approved the final draft.

## Data Availability

Our algorithm, implemented in R, is freely available at GitHub: https://github.com/wilkinsonlab/robust-clustering-metagenomics.

Our output data files are available at Zenodo: 10.5281/zenodo.1485916, including a set of graphs, and the <sample, state > annotations for downstream study, in a text file and a R phyloseq object.

## Supplemental Information

Supplemental information for this article can be found online at http://dx.doi.org/10.7717/peerj.6657#supplemental-information.

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
