# Peer review of "Robust and automatic definition of microbiome states"

_PeerJ, doi:10.7717/peerj.6657_

## Round 0.1 · original submission · Major Revisions

I have now obtained three reviews of your paper and all reviewers recommend 'major revision'. So that is also what I am recommending. All reviewers have commented that the paper really struggles determine what these sub-states mean and how they are really useful. Please pay particular attention to these comments and revise accordingly.

·

Basic reporting

Generally, the syntax of this manuscript needs to be improved. It is often casual in tone and rife with colloquial word choice. Interrupters, parenthetical statements, and transition statements (eg. ‘such as’, ‘here’, ‘for example’) are used too liberally. There are grammatical errors. The entire paper needs a heavy edit for professional language. In the spirit of Strunk and White, needless words should be deleted.

The materials and methods section is well written and clear.

Specific comments:
Lines 28-29: cavity does not describe where microbiomes are, yet there is a parenthetical statement regarding distinct environments directly following. This second statement should be used to describe the microbiome more generally.

Lines 33-34 sentences starting “Albeit…” and “Some studies…” are fragments and should be combined into a complete thought. The following statements 37-46 should be revised to fit with the earlier changes suggested in 33-34.

Line 60 contains an example of a distracting and non-contributing parenthetical statement.

Lines 238-240 are not needed, as the idea contained within is implicit in the following text.

Lines 328-329 contain an introductory sentence too casual for a manuscript.

Lines 367-370 contain a run-on sentence which obfuscates your main point.

Lines 390-392 contain further casual language.

Line 427, ‘an standard’ is a mistake and should be corrected.

The tables should both be reformatted and given proper legends. They are difficult to understand quickly and would benefit from better labels and graphic design. I personally do not like labelling the datasets by the author, but it would not prevent me from accepting the manuscript. However, ‘no.clusters’ and ‘no.taxa’ should be written out in English to prevent confusion. It is very frustrating that the headers contain back slashes and yet the cells contain forward slashes.

The figures should be labeled better – rather than describing where a panel is in the figure legend, please label it and use the legend to describe what the figure is, rather than where it is in space.

The conclusion section needs to be edited for brevity.

Experimental design

The logic of this study is very clear and I think that this protocol and algorithm fills an important knowledge gap / tool niche. As stated in the manuscript, enterotyping is often ad hoc and this may be a solution to the softer end of the science.

The methods section is very clear and well cited.

Validity of the findings

I am cautious to agree with the assertion that this algorithm ‘broadly applicable’ as stated in the abstract and implied in the body. While there are high value datasets included in the study, four sets is but a small fraction of the data extant. It would be helpful to further validate these findings in other diverse populations, other models or nonhuman systems before claiming broadness or generalizability.

I do, however, think that this algorithm will contribute to the conversation regarding microbial states, enterotypes, or gradients. Without standardization of workflow, it is certainly a challenge to assess change over time and differences between populations or localities.

The transparency of the description of the algorithm, specifically figure 1, should be commended. Its sometimes a challenge to reconstruct the logic of tool design and, here, the reader is shown the plan with great clarity.

Additional comments

The approach is interesting and needed to help resolve outstanding tension in the field as well articulated in the Patrick O’Toole’s 2012 comment (doi:10.1038/nrmicro2859) and the cited Costea//Bork perspective from 2017 (10.1038/s41564-017-0072-8). The point, however, is often muddied by poorly presented figures and tables and bad writing.

I would like to see this paper largely reconstructed for clarity and brevity in the case that the methods described within become part of the standards repertoire as you suggest in your discussion, lines 380-385. Reanalysis of Human Microbiome Project data would certainly help bolster the position that this is a generalizable tool. I recognize that it is no easy task to amass set after set of data and that this concern will be mitigated upon publication, adoption in the field, and iterative design. However, in this case more is better, or it should be deemed a proof of concept for a useful new standard.

Reviewer 2 ·

Basic reporting

Honestly, this manuscript looks like an unfinished work in progress and needs to be re-thought from the basic message to reporting. For example, I could not find references to Table 2 or Figure 3 in the text.

Experimental design

In my opinion, the best bet to get this work published is to reinvent is as an application note rather than original research. There's probably most value in the algorithm as a collection of working standard R code than in any of the results reported.

Validity of the findings

Claim: Generic, domain-independent and automatic procedure to determine a reliable set of microbiome sub-states within dataset and with respect to the conditions of the study.

The authors try to make a point about the difference between states and sub-states, however, the exact difference between the two escapes this reviewer, especially, given the fact that the methodology to define both of these are very similar. Further, the microbiome field is converging on the community state type (CST); perhaps, the authors could use more standard terminology here.

How do the authors define reliability? Although the manuscript defines some measures of robustness, it is never made explicit what it means for CSTs to be reliable.

What does domain-independent mean?

The methods presented are all unsupervised, therefore, I do not see how these approaches define CSTs with respect to conditions of the study. Unsupervised approaches may result in CSTs that are coincidentally related to conditions of a study, but there is no explicit guarantee.

It is not clear why the authors are focusing on longitudinal datasets. The analyses do not in any way exploit the fact that the data are longitudinal and/or sampled multiple times from the same individuals.

Methodologically, no novel approaches are proposed. The manuscript proposes to exploit best practices in clustering and uses a few established techniques that have appeared elsewhere in microbiome literature. Therefore, it is not clear what audience will find this information useful. Nonetheless, there is nothing wrong with the approaches used by the authors.

I could not follow the definition of Jaccard similarity clustering assessment.


The most important unaddressed question by this manuscript and in the field is the evidence for clustering. The evaluation is performed on k>=2 number of clusters; it is never checked whether k=1 is the optimal clustering.

Cluster interpretation results could benefit from formal statistical testing of association between the clusters and the putative experimental variable/factor.


"The score is the frequency of correct classification pairs." => "The score is the frequency of matching classification pairs." ??

·

Basic reporting

The english grammar is lacking.

It is difficult to understand how to use these "cluster" results. So what if they found two clusters -- what are they and how can we use this to analyze the results?

The description of the pipeline is vague.

In general, it is hard to understand the relevance of this paper because the authors do not explain how their results can be used.

Experimental design

Is agglom clustering the right approach for measuring the evolution of “substates” if the clusters are discrete in that the OTUs do not overlap from cluster to cluster? One would presume that a transition from substate 1 to substate 2 may involve shifts in the relative abundance of certain taxa that fall within a cluster and not a shift from OTUs specific to cluster x to OTUs specific to cluster y.

Just validating their results against the original papers is not enough.
The authors need to show the advantages/disadvantages of their approach by generating simulated data and see if they can capture the predefined substates, and then they need to describe at what point their approach fails and to what parameters their approach is sensitive to.

They also claim they want to describe the dynamics of these clusters – i.e., how they behave over time. This was not shown.

In Section MATERIALS AND METHODS, authors mentioned they used SI to determine best number of clusters. But the authors didn’t discuss the reason why this SI works well for this task expect for barely mentioned that it was from a previous research. It would be beneficial if the authors could provide some insights about why SI is helpful to find good number of clusters in microbiome dataset.

In Section MATERIALS AND METHODS, authors mentioned the threshold for PS is 0.8 and the threshold of Jaccard is 0.75. But the authors didn’t discuss the selection of threshold in this section.

In Section RESULTS, by looking at Figure 1, it seems that Jaccard similarity criterion is redundant as it is always better than PS. Did the author also evaluated the efficacy of the metrics used in the pipeline to avoid unnecessary computation?

Validity of the findings

Figure 2 needs to be redone. It’s too difficult to distinguish shapes. Decide to plot ellipses or to use facets.

Clustering samples from subject A and B together makes no sense for David data because we are not interested in how well we can separate two subjects. Instead, the goal of this paper is to identify sub-states that can capture transitions in longitudinal microbiome data. Clustering samples together will reduce each resolution because the intra class distance between samples from two subject A and B is too big.

Why only 2 clusters are found for David et al. (2014)? There is a pre-state, a dysbiosis state, and a post-state (Figure 3F of the original paper).. so there should be at least 3, which shows that this paper doesn't produce proper results. How about looking at Fig 4B in https://www.biorxiv.org/content/biorxiv/early/2017/08/15/176412.full.pdf which shows 5 clusters... with 3 of them being major, with corresponding taxonomic relative abundance and posterior probability of cluster over time? This is a much more in-depth analysis than just determining the number of clusters.

The authors previously mentioned that their pipeline can provide new insight into the data structure. But from the result, readers couldn’t really find interesting and novel explanations respect to the clusters the pipelines found. Basically, the pipeline can recover metadata information by the clusters to some extent, but the resolution is relatively low because there is no further analysis followed up by digging into each cluster and try to interpret the intrinsic relationship between clusters (OTU profiles that are similar, for example).

Additional comments

Line 66: cite https://genomebiology.biomedcentral.com/articles/10.1186/gb-2014-15-7-r89 and https://www.ncbi.nlm.nih.gov/pmc/articles/PMC5621509/

---

## Round 0.2 · Major Revisions

Unfortunately, both reviewers still have issues with your manuscript. Importantly, you failed to respond to a particular critique from Reviewer 3 from the previous round:

Just validating their results against the original papers is not enough.
The authors need to show the advantages/disadvantages of their approach by generating simulated data and see if they can capture the predefined substates, and then they need to describe at what point their approach fails and to what parameters their approach is sensitive to.

This reviewer reiterates this concern in this round of review. I agree. This is a standard for bioinformatics papers to do such a comparison. In fact, it would be quite easy to do in the section 'Relationship to other definitions of microbiome 'state'. You've compared results through a reanalysis of the Ravel et al. data, but you only include your method to compare different concepts of 'state'. Simply run these 'variety of clustering algorithms' to compare your results to other clustering approaches as well. This will require additional effort, but it is standard for such a paper. Likewise, as reviewer 1 points out, a link to software availability is also standard. I believe that is part of the PeerJ submission process and it should be included in the manuscript so readers can find the software and actually use it. If you are not willing to perform the comparison, then you should withdraw the paper. Otherwise, it seems pretty straight forward to accommodate this and the additional concerns.

·

Basic reporting

Dear authors,

This is such a vast improvement on the previous manuscript. Thank you for your careful work and heavy edits - it really clarifies your contributions documented in this manuscript. There appears to be a very minor line splice over lines 481-484 where something may have been transposed from your clipboard. The sentence will make more sense once this is corrected.

Otherwise, all comments have been addressed or rewritten.

Experimental design

The design of the experiments is repetitive and clear and spans a broad swath of both microbiomes and the mathematical evaluation of their emergent clusters. I am satisfied that they indeed cluster microbiomes in a data driven fashion. I don't think you include a link to your github - obviously you are findable, but including a link to the code would be helpful / thorough.

Validity of the findings

This manuscript seems to have improved significantly, and for the better, with revision. Based on the information provided in the manuscript and the supplemental files I believe that their approach provides biology-agnostic, data driven clustering. Based on their github repository, I see the r scripts to be well documented and clearly useful to the microbiome computational biology community.

Additional comments

Thank you for taking all three reviewers comments so seriously. You took a good idea and made it into a clear paper. The only reason that I have not accepted this paper outright is because of the presumed typo above and my hope that the editor will suggest a place where it is appropriate to link your github or r scripts. I wish you the best in all your future endeavors.

Sincerely,
Jonathan LoTempio Jr

·

Basic reporting

The manuscript was drastically changed and hard to compare with the previous version and almost warrants a resubmit.

"There are relatively few studies about temporal dynamics within any microbiome site" -- I feel like the authors are leaving out numerous studies. Off the top of my head, I know this one: https://www.nature.com/articles/ncomms5344

Their approach -- clustering and optimally determining the number of clusters based on different distance measures etc. -- does not seem very novel. And they fail to really compare their methods and results to other papers' methods and results. Like, we had you site Ananke (that and other methods exist)... how does your method compare, even if it does something different -- compare and contrast to other algorithms.

Experimental design

A state should be defined for the input data and desired output. The following definition is confusing: "In this manuscript, we define a microbiome ”state” as a collection of constraints satisfied by the58microbiota of some given samples, that are not satisfied by other samples, allowing them to be reliably59distinguished from one another." -- So, is a state a group of samples or is it a group of OTUs? This explanation does not help: "Therefore, our approach to defining states, described in detail below, consists100of applying a clustering algorithm to the OTU data, taking a metagenomics beta diversity metric as the101distance measure between samples." ... sounds like states are clustering of samples.

The authors use k from 2 to 10 in their algorithm as the number of clusters. However, since you are working on longitude data, there is a chance that there are more than 10 changes in a long period of time. Therefore, it makes more sense to increase k until the clustering assessment scores stays the same or drops (thus you could have a way of smartly choosing k).

Your algorithm assumes that there are at least 2 clusters in the data, what if there is only 1 cluster?

For the David et al. data: As the authors wrote in their responses when they apply their algorithm to subject A only, they could find 2 clusters: “for subject A we get 2 clusters, that is in agreement with Figure 3B and 3D of the original study, with one cluster for samples during travel (approx. 70-120 collection days) and other cluster for the remaining samples”. Whereas in the manuscript, the authors just show the results of clustering A and B together and in this setting, the authors can’t find the cluster that correspond to A’s travel period. I still suggest the authors show the results of their clustering experiment on subject A and B separately in the paper.

In line 358-370, the authors talk about the composition of the clusters they found. I suggest to provide a figure to illustrate the composition. For example, a phylum level distribution for each cluster. They can also integrate this in their algorithm to help users to explore the clusters produced.

Validity of the findings

I am glad the authors brought up their recently published paper. I think the algorithm presented in this paper is a parameter sweep tool to find the best number of cluster because no further analysis is followed by the algorithm hence this algorithm itself can’t really identify biological meaningful states. However, when this algorithm is followed by the other method proposed in the authors’ recently published paper, we can identify interesting clusters as well as understand their transitions. That makes the clusters valuable and interpretable.

The authors should have included this algorithm in the previous paper in the methods. Without comparing this algorithm to other algorithms, this cannot be a standalone paper.

Additional comments

The writing is still poor, e.g. "'habits' such as lifestyle."

---

## Round 0.3 · accepted · Accept

Thank you for your careful response to the previous round of reviewing. I feel you have responded well and adjusted your manuscript accordingly. Therefore, I am now happy to recommend acceptance.

#